# RAD-ical New Insights into RAD51 Regulation

**DOI:** 10.3390/genes9120629

**Published:** 2018-12-13

**Authors:** Meghan R. Sullivan, Kara A. Bernstein

**Affiliations:** Department of Microbiology and Molecular Genetics, UPMC Hillman Cancer Center, University of Pittsburgh School of Medicine, 5117 Centre Avenue, Pittsburgh, PA 15213, USA; mrs149@pitt.edu

**Keywords:** homologous recombination, RAD51 paralogs, BRCA1, BRCA2, RAD51B, RAD51C, RAD51D, XRCC2, XRCC3, Shu complex, cancer

## Abstract

The accurate repair of DNA is critical for genome stability and cancer prevention. DNA double-strand breaks are one of the most toxic lesions; however, they can be repaired using homologous recombination. Homologous recombination is a high-fidelity DNA repair pathway that uses a homologous template for repair. One central HR step is RAD51 nucleoprotein filament formation on the single-stranded DNA ends, which is a step required for the homology search and strand invasion steps of HR. RAD51 filament formation is tightly controlled by many positive and negative regulators, which are collectively termed the RAD51 mediators. The RAD51 mediators function to nucleate, elongate, stabilize, and disassemble RAD51 during repair. In model organisms, RAD51 paralogs are RAD51 mediator proteins that structurally resemble RAD51 and promote its HR activity. New functions for the RAD51 paralogs during replication and in RAD51 filament flexibility have recently been uncovered. Mutations in the human RAD51 paralogs (RAD51B, RAD51C, RAD51D, XRCC2, XRCC3, and SWSAP1) are found in a subset of breast and ovarian cancers. Despite their discovery three decades ago, few advances have been made in understanding the function of the human RAD51 paralogs. Here, we discuss the current perspective on the in vivo and in vitro function of the RAD51 paralogs, and their relationship with cancer in vertebrate models.

## 1. Introduction to Double-Strand Break and Repair

Exogenous and endogenous DNA damage is constantly challenging our genomic integrity. Exogenous DNA damaging agents, such as radiation, ultraviolet light, and chemicals, or endogenously generated DNA damage, such as errors in replication or cellular processes that generate reactive oxygen species, create a wide variety of DNA lesions [1]. Maintaining genome stability requires many coordinated processes within the cell to ensure the conservation of our genetic material through each cell division [2]. Collectively, these processes are referred to as the DNA damage response (DDR; See Table 1 for all of the abbreviations used) [3]. Accurate DNA repair is a key part of the DDR, and its loss leads to genome instability, which is a hallmark of cancer development [4]. The most deleterious type of DNA damage is a double-strand break (DSB), as even a single unrepaired DSB results in cell death [5], and the misrepair of DSBs is associated with increased genomic instability and consequently tumorigenesis [2]. Therefore, cells have evolved highly specialized responses to recognize and repair DSBs.

Non-homologous end joining (NHEJ) and homologous recombination (HR) are the two predominant pathways to repair DSBs [6]. Once a DSB is recognized and the DDR is initiated, the cell must determine the appropriate repair pathway based on the nature of the break and the availability of a potential repair template. A major factor governing repair pathway choice is the cell cycle phase. The predominant DSB repair pathway in human cells is NHEJ; it can be used throughout the cell cycle, especially in the G1 phase [6]. Conversely, HR is primarily restricted to S and G2 phases when a sister chromatid is available as a repair template. Repair using NHEJ does not rely on a homologous template, but rather employs minimal processing around the break site and ligation of the ends [7]. Therefore, NHEJ more frequently results in a loss of genetic material through microinsertions and deletions (INDELs) [7]. In contrast, HR is a tightly regulated and faithful template-guided repair process that replaces the lost or resected DNA around the damage using the information provided by an intact homologous sequence such as a sister chromatid or homologous chromosome. This increased fidelity ensures the preservation of the genome through each cell division. Commitment to HR requires DSB end resection and the formation of RAD51 nucleoprotein filaments.

### 1.1. Commitment to Homolgous Recombination through Double-Strand Break End Resection and RAD51 Filament Formation

The HR pathway initiation requires 5′ to 3′ end resection at the break site, exposing single-stranded DNA (ssDNA) overhangs that ultimately prevent canonical NHEJ from repairing the break (Figure 1) [6]. DNA end resection is initiated by MRE11–RAD50–NBS1 (MRN) binding to the DNA ends, which subsequently recruits CtIP to generate 3′ single-stranded DNA (ssDNA) overhangs [6,8]. In addition to the resection activity of MRE11, repair pathway choice is also directed by the opposing actions of 53BP1 and BRCA1 [9,10]. Once a break is detected, 53BP1 and BRCA1 compete for directing the cell to commit to NHEJ or HR, respectively (Figure 1A) [10,11]. 53BP1 promotes NHEJ by inhibiting DNA end resection while simultaneously tethering two double-stranded DNA (dsDNA) ends together, enabling their subsequent ligation [11]. How BRCA1 inhibits 53BP1 activity remains unclear. However, when BRCA1 binds BARD1, it can ubiquitinate CtIP, increasing CtIP affinity for DNA and thus promoting resection [11,12,13]. At the same time, the DNA ends are shielded from resection through the 53BP1-interacting partners, RIF1 and the newly identified shieldin complex (REV7–SHLD1–SHLD2–SHLD3) [14,15,16,17]. The loss of 53BP1 or the shieldin complex impairs NHEJ, resulting in increased HR (Figure 1A) [9,10,14,15,16,17]. Preventing extensive end resection is important for limiting hyperrecombination by HR and preventing the loss of genetic material. Extensive resection can result in loss of heterozygosity by alternative deleterious repair pathways such as single-strand annealing (SSA; Figure 1B) or break-induced replication (BIR; not pictured) [18].

Once DNA end resection occurs, the HR pathway can be utilized. Increasing evidence suggests that the primary role of HR is actually to repair DNA damage that occurs during replication [2]. While canonical HR repairs a direct DSB, this pathway can also repair lesions produced by stalled or collapsed replication forks [2]. Cells commit to a homology-directed mechanism of repair when extensive resection is performed by the action of multiple nucleases. Short and long-term resection is mediated by MRN/CtIP in conjunction with EXO1 or BLM and DNA2 (Figure 1B) [19]. This resection reveals 3′ ssDNA ends, which are quickly coated by the replication protein A (RPA) complex [19,20]. Filaments coated by RPA ensure that the ssDNA overhangs are not degraded, and prevent secondary structures from forming [19]. RAD51 then displaces RPA to form the pre-synaptic filament, and this requires the activity of several so-called RAD51 mediator proteins (Figure 1C) [21]. RAD51 nucleoprotein filaments search for a homologous sequence to invade and displace one strand of the homologous template to form a displacement loop (D-loop; Figure 1D) [22]. In canonical HR, this structure allows for the pairing of the broken strand with the displaced strand to form a heteroduplex (Figure 1D), and DNA synthesis restores any missing nucleotides at the break site (Figure 1E). Subsequently, second end capture results in the formation of a double Holliday junction (dHJ) (Figure 1G). This intermediate is resolved through either a dissolution or resolution mechanism, yielding noncrossover (NCO) [23] or crossover (CO) (Figure 1H,I) [22]. Alternatively, during synthesis-dependent strand annealing (SDSA), only one-end invasion occurs, thus forming a single Holliday junction, and this intermediate is dissolved into an NCO product (Figure 1F) [22]. In this review, we will focus on the mechanisms that regulate the commitment to HR through the RAD51 mediators. 

### 1.2. Overview of RAD51 Structure, Function, and Activity

RAD51 filament formation is key for commitment to HR and is a highly conserved step [21]. The centrality of RAD51 to HR is underscored by its evolutionary conservation from bacterial RecA to human RAD51, as well as by the amino acid sequence similarity to its meiotic counterpart, DMC1, and its paralogs (described in Section 1.3) [24].

RAD51 plays a critical role during DNA homology search and strand invasion. RAD51 assembles into a heptamer that encircles the DNA, forming a helical nucleoprotein filament in which one RAD51 molecule binds to three nucleotides [25,26]. This nucleoprotein filament is termed the presynaptic filament [27]. Recently, total internal reflection fluorescence (TIRF) microscopy revealed that a homology search by the presynaptic filament efficiently samples dsDNA in at least eight nucleotide increments, discounting matches of seven or fewer nucleotides [25]. TIRF microscopy further revealed that after finding eight nucleotide microhomologies, RAD51-mediated strand exchange occurs in three nucleotide steps with proper Watson–Crick base pairing, and is conserved from bacterial RecA to human RAD51 [26]. The homology search has been modeled in vivo in budding yeast using fluorescent microscopy and endonuclease-induced DSBs [27]. Upon DNA damage, both the broken ends and undamaged chromosomes increase their local mobility to enable the search for homology in both haploid and diploid yeast cells [28,29]. In vivo, the homology search and movement of the DNA ends require budding yeast Rad51, DNA end resection proteins such as Sae2/CtIP, and the DNA damage checkpoint (Mec1, Rad9, and Rad53) [28,29]. In human cells, movement of the DSB ends during the homology search is controversial. In some cases, DSB mobility was not observed when monitoring both ends of a break site in live cells [30]. This immobility is thought to be important for preserving genome stability by preventing illegitimate fusions [31]. In other instances, the movement of DSB ends was observed in live cells through the clustering of chromosome domains [32,33]. DSBs at telomeres also exhibit increased mobility, which results in telomere–telomere recombination [34]. RAD51 has proved challenging to study in mammalian systems because, unlike yeast, the loss of functional RAD51 is not tolerated in mouse models, and RAD51^-/-^ cells cannot be propagated [35,36]. Therefore, much of what we know about the homology search is based upon work done in model organisms such as yeast.

### 1.3. Overview of Key RAD51 Regulators

Although the RAD51 nucleoprotein filament is required for the homology search and strand invasion steps of HR, a rate-limiting step to RAD51 filament formation is the displacement of RPA from ssDNA [22]. Since RPA binds with higher affinity to ssDNA than RAD51 [37,38], RAD51 mediator proteins are required to assemble RAD51 on ssDNA by nucleating, elongating, and stabilizing the RAD51 nucleoprotein filament. In addition, new roles for these proteins have been identified in RAD51 filament flexibility and even end capping, which is thought to stimulate strand exchange [39,40].

In mammalian cells, RAD51 filament nucleation is mediated by the RAD51 loader, BRCA2, and this function is carried out in other eukaryotes, such as budding yeast, by the Rad52 protein [41,42,43]. In humans, RAD51 binds BRCA2 through the BRC repeats and the C-terminal domain of BRCA2 [43,44,45,46]. The BRC repeats in BRCA2 mimic the oligomerization interface of RAD51, thus enabling RAD51 loading [45,47]. BRCA2 delivers RAD51 monomers to ssDNA rather than dsDNA, allowing filament formation and ultimately promoting RAD51 strand-exchange activity [43,47,48]. Underscoring the importance of the coordination of RAD51 activity by BRCA2, BRCA2 mutations in the BRC repeats have been found in cancers. Furthermore, mice with deletion of the exon containing the BRC repeat of BRCA2 are inviable [49,50,51]. BRCA2 also binds and coordinates the activity of several other recombination factors, including DSS1 and PALB2 to promote RAD51 loading and activity [42].

In addition to BRCA2, other RAD51 mediators include the RAD51 paralogs, which are proteins that structurally resemble RAD51 itself, and the Shu complex, which is a RAD51 paralog-containing complex. RAD51 paralogs arose from a gene duplication of the ancestral RADA protein in archaea and have maintained their structural similarity to RAD51. In humans, there are six RAD51 paralogs including RAD51B, RAD51C, RAD51D, XRCC2, XRCC3, and SWSAP1. Once RAD51 filaments are nucleated, the RAD51 paralogs are thought to aid in several aspects of RAD51 regulation such as the stabilization and elongation of the RAD51 filament itself and in filament remodeling to facilitate the homology search. However, the precise function of the individual RAD51 paralogs remains largely enigmatic, and why so many RAD51 paralogs are needed is unknown.

In addition to the RAD51 paralogs, other positive RAD51 regulators also aid in downstream recombination steps after RAD51 filaments have formed. The DNA translocase paralogs RAD54A and RAD54B are chromatin remodelers that enable strand exchange through their dsDNA-dependent ATPase activities [52,53,54]. In addition to the positive regulators, negative regulators also aid in the disassembly of RAD51 filaments. FBH2 and RECQL5 act as antirecombinases to dissociate RAD51 from ssDNA, whereas FANCM and RTEL specifically function during D-loop disassembly [54,55,56]. These negative regulators are equally important, as they modulate recombination by limiting RAD51 activity at illegitimate recombination sites (Figure 1C,D) [54].

## 2. RAD51 Regulation in Mammalian Models

### 2.1. RAD51 Paralog Containing Complexes

The mammalian RAD51 paralogs were first identified 30 years ago [57]. Five of the six RAD51 paralogs are considered canonical RAD51 paralogs (RAD51B, RAD51C, RAD51D, XRCC2, XRCC3) and share 20–30% amino acid sequence identity with RAD51 itself, particularly around the Walker A and B motifs [58,59,60,61]. The more recently identified SWSAP1 protein is a highly divergent RAD51 paralog that shares its closest sequence homology with RadA (~24%), which is an archaeal RecA family member and also contains Walker A and B motifs [62]. The RAD51 paralogs assemble into subcomplexes in vivo as the heterotetramer BCDX2 (RAD51B, RAD51C, RAD51D, and XRCC2), the heterodimer CX3 (RAD51C and XRCC3), and the Shu complex (SWSAP1 and SWS1) (Figure 2) [62,63,64,65]. RAD51C is also a member of an additional complex that consists of BRCA2, PALB2, and RAD51 itself (Figure 2). Unlike the canonical RAD51 paralog subcomplexes, the Shu complex consists of a Shu2/SWS1 protein family member that is characterized by a conserved zinc-finger like binding motif, the SWIM domain, CXC…Xn…CXHXXA, where X is any amino acid. In all of the organisms where the Shu complex has been analyzed, the Shu2/SWS1 protein family member interacts with the RAD51 paralogs to regulate RAD51 function [66,67]. All of the RAD51 paralog-containing complexes are thought to promote RAD51-mediated activities, although their precise composition and function in this process is largely unknown.

Initial determination of RAD51 paralog subcomplex assembly was determined using yeast two-hybrid and yeast three-hybrid systems due to the insolubility of recombinantly expressed RAD51 paralogs [62,64,68,69,70]. Although biochemical investigation of the mammalian RAD51 paralogs has lagged significantly behind cellular studies, the protein–protein interactions of the RAD51 paralog subcomplexes, BCDX2 and CX3, were later confirmed using recombinant proteins purified from Escherichia coli and Sf9 insect cells [63,64]. The RAD51 paralogs are thought to be incorporated into their respective subcomplexes in a 1:1 stoichiometry. For example, recombinant CX3 and BCDX2 assemble into a 1:1 and 1:1:1:1 stoichiometry, respectively, when purified from either insect Sf9 cells or human HeLa cells [63,64,71]. In vitro, the BCDX2 complex can also assemble into stable heterodimers, which include BC (RAD51B–RAD51C subcomplex) and DX2 (RAD51D–XRCC2 subcomplex) [72,73]. Unlike RAD51, which interacts with additional RAD51 monomers to form a filament, the RAD51 paralogs do not. Instead, they assemble into heterodimers, and these protein–protein interactions are critical for their stability [62,74]. Within the BCDX2 complex, RAD51C and RAD51D interact with each other, and also with RAD51B or XRCC2, respectively (Figure 2) [64,68,73,75]. This is similar to the budding yeast Shu complex, which forms a horseshoe shape, as revealed by X-ray crystallography [76].

Recently, RAD51C was found to act in a third complex that is distinct from the other RAD51 paralog subcomplexes. Mass spectrometry revealed that RAD51C interacts directly with PALB2, which acts as a scaffold to simultaneously bind RAD51, RAD51C, and BRCA2 in HeLa S3 cells [77]. PALB2, as the “partner-and-localizer of BRCA2”, is necessary to recruit BRCA2 to sites of DNA damage [78]. PALB2 mutants that disrupt RAD51C interaction show increased BRCA2 foci, but decreased RAD51 foci [77]. Therefore, the PALB2–RAD51–RAD51C–BRCA2 complex may facilitate BRCA2 removal after RAD51 filament nucleation [77].

### 2.2. In Vitro Characterization of RAD51 Paralog Function in RAD51 Pre-Synaptic and Post-Synaptic Filament Assembly

The initiation of RAD51 filament assembly on ssDNA overhangs may be facilitated, in part, by both the CX3 and BCDX2 complexes. Consistent with this notion, CX3 exhibits ATP-independent DNA binding affinity for ssDNA [63,71]. CX3 also binds to other DNA substrates such as 5′ or 3′ tailed DNA, but with reduced affinity [63,71] and has the lowest affinity for dsDNA [63,71]. Interestingly, the CX3 complex promotes DNA aggregation, which is suggestive of a role in annealing complementary DNA during the homology search of RAD51 filaments [63,71]. Together, this data suggests that CX3 may have an early function in RAD51 filament assembly. Similarly suggesting a role of the BCDX2 complex in RAD51 filament assembly, the BCDX2 complex exhibits a modest ATPase activity in the presence of ssDNA, but not in the presence of 5′ or 3′ tailed DNA or dsDNA [64]. Further supporting a role for the BCDX2 complex in RAD51 filament assembly, electron microscopy analysis has shown the BCDX2 complex bound to both ssDNA, as well as gaps and nicks in duplexed DNA [64]. Given that the RAD51 paralogs contain Walker A and Walker B motifs that are used for ATP binding, their role in RAD51 filament mediation may utilize ATP hydrolysis. For example, the BC heterodimer (RAD51B–RAD51C subcomplex) or the DX2 heterodimer alone (RAD51D–XRCC2 subcomplex) also binds ssDNA, and this binding stimulates ATPase activity [72,73]. Furthermore, the DX2 heterodimer ssDNA binding is enhanced upon ATP addition [72,79]. In addition to biochemical studies, the roles of BCDX2 and CX3 in filament formation are most strongly supported by the cellular studies discussed below [74,80]. Additionally, work with the yeast RAD51 paralogs support this pre-synaptic role. In vitro analysis has revealed that both the yeast RAD51 paralog-containing complexes, RAD55–RAD57 and the Shu complex, promote RAD51 pre-synaptic filament assembly [81,82]. How the RAD51 paralogs mechanistically aid in RAD51 filament assembly is still unknown (Figure 3). It has been hypothesized that perhaps they can either intercalate into the filament (Figure 3A; left side) or even form a cofilament that enables RAD51 elongation after BRCA2-mediated nucleation (Figure 3A; right side). For example, DX2 and CX3 were observed to form filament structures on ssDNA, although these structures significantly differed from RAD51 nucleoprotein filaments [71,79]. Alternatively, the RAD51 paralogs could potentially cap the DNA ends to prevent RAD51 filament disassembly, which is similar to what has been observed in yeast for Rad55–Rad57 (Figure 3B) [62]. Lastly, it has been hypothesized that different RAD51 paralog-containing complexes may promote HR depending upon the nature of the DNA lesion, particularly for the Shu complex [66,67,83,84]. For example, the Shu complex shows specificity for promoting tolerance of DNA damage, such as an abasic site, in a replication-specific context [83].

Conflicting biochemical evidence has also described post-synaptic roles for the RAD51 paralogs as well. For example, contradictory evidence suggests that CX3 either does or does not aid RAD51-mediated D-loop formation [63,71]. Suggesting a role for DX2 and BC in strand exchange, the DX2 heterodimer also catalyzes homologous pairing, enabling D-loop formation [79], while the BC heterodimer enhances RAD51-mediated strand exchange in the presence of RPA [73]. It is possible that the incorporation of the RAD51 paralogs into the RAD51 filament could change the conformation of the RAD51 filament to enable increased flexibility for strand exchange, as well as promote filament disassembly to allow the subsequent steps of HR to proceed (Figure 3B). Work with the *Caenorhabditis elegans* RAD-51 paralogs, RFS-1 and RIP-1, have provided the most convincing biochemical evidence for the worm RAD-51 paralogs’ role in increasing filament remodeling [39,40]. The authors used stop flow experiments and cryogenic electron microscopy to show that the worm RAD-51 paralogs facilitate a conformation that enables base pairing and strand exchange. They propose a model in which BRC-2 nucleates RAD-51 displacing RPA, and the RAD-51 paralogs stabilize and remodel the pre-synaptic filament. The RAD-51 paralogs change RAD-51 pre-synaptic filament conformation by capping the 5′ end and remodeling up to 40 nucleotides of the 5′–3′ filament [39,40]. These RAD-51 paralog activities are dependent on nucleotide binding, but not ATP hydrolysis [39,40]. Beyond these initial characterizations, more detailed in vitro studies with the human RAD51 paralogs are lacking. For example, the human RAD51 paralogs have not yet been purified individually, nor have their crystal structures been determined. Therefore, most of our current understanding of RAD51 paralog function comes from molecular studies in model organisms that have addressed the steps of repair at which the RAD51 paralogs act.

### 2.3. In Vivo Characterization of RAD51 Paralog Function in Vertebrates

#### 2.3.1. RAD51 Paralog Knockout Mice and Mouse Embryonic Fibroblasts

Since their initial discovery, technical challenges have limited the study of the RAD51 paralogs in vivo [21,58]. For example, mouse knockout models for the five canonical RAD51 paralogs result in embryonic lethality (summarized in Table 2). Supporting unique functions for each RAD51 paralog, the knockout models arrest at different developmental stages (Table 2); *Rad51b^-/-^* (E7.5–E8.5), *Rad51c^-/-^* (E8.5), *Rad51d^-/-^* (E9.0–E10.0), and *Xrcc2^-/-^* (E10.5, died at birth) [86,87,88,89]. This embryonic lethality mirrors that of *Brca2^-/-^* knockout mice (~E8–E9), and provided early evidence that the RAD51 paralogs, such as BRCA2, have important HR and developmental functions [90]. Recently, the highly divergent RAD51 paralog SWSAP1 and its binding partner SWS1 were shown to produce viable, but sterile, knockout mice [84]. The sterility observed is due to defects in RAD51-mediated and DMC1-mediated meiotic recombination. These mouse models provide new opportunities to examine RAD51 paralog function, which has not been possible with the canonical RAD51 paralogs.

In addition to a lack of animal models for the canonical RAD51 paralogs, creating mouse embryonic fibroblasts (MEFs) from these knockout embryos has been challenging. Suggesting *RAD51C* and *RAD51D* are essential, MEFs could not be derived from *RAD51C^ko/ko^* or *RAD51D^ko/ko^* mice, and conditional *RAD51C* knockout MEFs could not be propagated [87,88]. In contrast to *RAD51C^ko/ko^* and *RAD51D^ko/ko^*, *XRCC2^ko/ko^* MEFs were created and found to exhibit fewer RAD51 foci following ionizing radiation-induced DNA damage, and increased mitomycin C (MMC) sensitivity with fewer sister-chromatid exchanges [91]. Most intriguingly, even a *XRCC2^ko/+^* heterozygote knockout displayed genetic instability [91]. This result has important clinical implications for *XRCC2* mutation carriers.

Interestingly, *Trp53* knockout slightly extended the embryonic development of *RAD51B*, *RAD51C*, and *RAD51D* knockout mice (Table 2) [87,89,92]. The greatest rescue is observed with *XRCC2* knockout mice, where *Trp53* knockout extended development by six days [61]. These results are particularly interesting in the context of ovarian cancer, where RAD51 paralog germline and somatic mutations are found in p53-deficient tumors [93,94]. In this context, p53 disruption could enable growth with RAD51 paralog deficiency. Although mouse models result in embryonic lethality, MEFs have been derived from three of the RAD51 paralog knockout mice in a p53-deficient background (*RAD51C^ko/ko^*/*Trp53^ko/ko^*; *RAD51D^ko/ko^*/*Trp53^ko/ko^*; XRCC2*^ko/ko^*/*Trp53^ko/ko^*) [61,87,92,95]. These MEFs exhibit defects that are consistent with decreased RAD51 loading or activity (summarized in Table 2). For example, *RAD51D^ko/ko^*/*Trp53^ko/ko^* MEFs have decreased MMC-induced sister chromatid exchanges (SCEs), which result from RAD51-mediated crossover events [92]. This is further supported by a decrease in RAD51 foci formation after irradiation (IR) in both *RAD51C^ko/ko^*/*Trp53^ko/ko^* and *RAD51D^ko/ko^*/*Trp53^ko/ko^* MEFs [87,92]. These RAD51 paralog-deficient MEFs are chromosomally unstable with increased chromatid breaks, gaps, and exchanges [87,92]. In addition to genetic instability, RAD51 paralog disruption in combination with p53 results in extreme sensitivity to the DNA crosslinking agent MMC [61,87,92,95]. Severe sensitivity to crosslinking agents is a defining feature of cells derived from Fanconi anemia (FA) patients and unsurprisingly, RAD51C (FANCO) and XRCC2 (FANCU) mutations have been uncovered in FA or FA-like patients [96,97,98,99].

#### 2.3.2. RAD51 Paralog Knockout Hamster, Chicken, and Tumor Cell Lines

The most progress in understanding the role of mammalian/vertebrate RAD51 paralogs has come from studies in Chinese hamster ovary (CHO) and chicken (DT40) cell lines where the RAD51 paralogs are not essential for survival [57,80,100,101,102,103]. Both CHO and DT40 cells have mutant p53, which is likely enabling the deletion of RAD51 paralogs to be tolerated in culture [102,104]. CHO cells lacking XRCC2 (XRCC2−/−, irs1: ionizing radiation sensitive 1) have the greatest sensitivity to MMC, but are also sensitive to ionizing radiation (IR), ultraviolet light, and ethyl methanesulfonate [57,58]. Chinese hamster ovary cells lacking XRCC3 (XRCC3−/−, irs1SF) also exhibit increased sensitivity to IR and fail to form RAD51 foci after damage [58,103,105]. Similar to CHO cells, DT40 (derived from B-lymphocytes) cells are p53 deficient, and tolerate a loss of any of the five canonical RAD51 paralogs [80,102]. DT40 RAD51 paralog knockouts exhibit genomic instability, as revealed by spontaneous chromosomal breaks, a reduction in MMC-induced SCEs, and decreased IR-induced RAD51 foci [80,102]. Furthermore, each RAD51 paralog DT40 knockout cell line shows increased sensitivity to DNA-damaging agents such as IR, MMC, and cisplatin [80,102]. These DT40 cell lines have been complemented with human or mouse cDNA, which rescued genome instability phenotypes in RAD51B−/−, RAD51D−/−, XRCC2−/−, and XRCC3−/− [80,102]. Interestingly, RAD51C-/- DT40 could not be complemented with human RAD51C, and this has limited RAD51C functional analysis [80]. Since their initial characterization, these cell lines have complemented and confirmed the results obtained from the mouse models.

In contrast to other vertebrate models, knockdown of RAD51 paralogs in human cancer cell lines (HeLa, HT1080, MCF7, and U2OS) has been challenging. For example, small interfering RNA (siRNA) depletion of any individual RAD51 paralog destabilizes its binding partners (i.e., siXRCC2 decreases the expression of endogenous RAD51D), and siRNAs have variable levels of knockdown efficiency [74,106]. RAD51C is particularly problematic, as it is a member of both the BCDX2 and CX3 complexes, and its depletion destabilizes members of both subcomplexes [106]. The stability dependency between the RAD51 paralogs makes understanding their unique contributions particularly difficult. Furthermore, knockdown of RAD51C is highly toxic to HeLa cells, as measured by plating efficiency, and delays cell cycle progression from the G1 phase into the S and G2 phases [106].

The importance of each human RAD51 paralog in HR has been demonstrated in MCF7 and U2OS cell lines by measuring HR following an endonuclease-induced DSB and by monitoring cells for DNA damage sensitivity [74,106]. Consistent with a HR function, the elevated IR-sensitivity of RAD51C-depleted HeLa cells was specific to the S/G2 cell cycle phase when HR is most active, while G1 phase cells were equally sensitive as controls when other repair pathways predominate [106]. Furthermore, increasing evidence suggests that the RAD51 paralog subcomplexes likely have non-overlapping roles. For example, siRNA knockdown of RAD51D, which disrupts the BCDX2 complex specifically, results in decreased RAD51 foci formation following IR exposure, whereas siRNA knockdown of XRCC3, which disrupts the CX3 complex specifically, does not [74]. Concurrent siRNA knockdown of both RAD51B and RAD51D does not further impair RAD51 foci formation upon IR treatment [74]. These results suggest that the CX3 and BCDX2 complexes function independently. Furthermore, BCDX2 likely acts upstream of RAD51, whereas the CX3 complex may function after RAD51 filament formation [74]. This result is contradictory to in vitro work suggesting that CX3 has an early function in RAD51 filament assembly [63,71]. Suggesting that BRCA2 recruitment is independent of the RAD51 paralogs, the depletion of either BCDX2 or CX3 does not impair BRCA2 foci formation after IR [74]. Altogether, these cell-based studies support the notion that the RAD51 paralogs might function during both pre-synaptic and post-synaptic filament assembly (Figure 3).

### 2.4. The RAD51 Paralogs Function at Replication Forks

The RAD51 paralogs also play critical roles at damaged replication forks [85]. RAD51 paralog disruption leads to sensitivity to genotoxic agents that cause replication-associated damage such as the alkylating agent methylmethane sulfonate (MMS) [87,107]. Alkylation damage is primarily repaired through the base excision repair pathway (BER); however, if the replication fork encounters a BER DNA processing intermediate, these DNA intermediates can slow or even collapse replication forks [108]. Similar to agents that directly induce DSBs, both *RAD51C^ko/ko^/Trp53^ko/ko^* and *RAD51D^ko/ko^/Trp53^ko/ko^* MEFs are sensitive to MMS (Table 2) [87,107]. Similarly, knockdown of the human Shu complex members, SWS1 or SWSAP1, also increases MMS sensitivity and reduces RAD51 foci formation [62,65]. Interestingly, the Shu complex function is specific to these types of lesions, as SWS1 or SWSAP1 knockdown cells do not exhibit sensitivity to IR [62]. This is consistent with the role for the yeast Shu complex in tolerance of MMS-induced DNA damage during S phase [83]. 

In mammalian cells, emerging new roles in replication fork protection and restart have been identified for other RAD51 mediators such as BRCA2 [109,110]. Fiber-spreading methods have revealed that BRCA2 protects replication forks from MRE11-mediated degradation [109]. RAD51 has also been implicated to have a role in fork protection [111]. Direct evidence for a role of the human RAD51 paralogs at replicative damage has only recently been investigated [85]. A study by Somyajit et al. examined the consequences of loss of three RAD51 paralogs, RAD51C, XRCC2, and XRCC3, using RAD51 paralog CHO mutant cell lines and HeLa cell knockdowns [85]. Suggesting that RAD51C, XRCC2, and XRCC3 protect replication forks, RAD51 paralog-deficient cells have increased MRE11-mediated degradation of nascently replicated DNA, which is observed by DNA fiber spreading (Figure 3C). This implicates both the BCDX2 and CX3 complexes as being involved in replication fork protection [85]. Unlike XRCC2, CX3 is uniquely important for replication fork restart, and this activity depends on their Walker A motifs [85]. This result suggests that the RAD51 paralog subcomplexes may play unique functions during the repair of replication damage [85].

## 3. RAD51 Mediators and Disease

Since HR is a high-fidelity DSB repair mechanism, mutations in HR genes are particularly deleterious to cells. The importance of maintaining this repair pathway is highlighted by the link between mutations in HR genes and several cancer-associated genetic diseases. Defects in HR genes cause many genetic syndromes, such as ataxia telangiectasia, Nijmegen break syndrome, FA, and Bloom’s syndrome [112]. Specifically, the RAD51 mediators and their interaction partners are heavily correlated to diseases defined by genomic instability that predispose individuals to cancer.

### 3.1. Genetic Syndromes Linked to RAD51 Mediators

While defects in HR genes are linked to several genetic syndromes, the RAD51 mediators are most closely associated with FA [113]. Fanconi anemia affects many systems of the body, causing bone marrow failure, anemia, congenital abnormalities, and cancers, amongst other clinical conditions [113]. Most notably, these patients have an early predisposition to several cancers of the blood, bone marrow, and solid tumors, which vary by complementation group [113,114]. Fanconi anemia is diagnosed by a sensitivity to the crosslinking agent MMC or diepoxybutane, resulting in chromosomal breaks and radials [99,113]. This MMC sensitivity is due to defects in interstrand crosslink (ICL) repair, either in the removal of the crosslink itself or in downstream HR steps [113]. FA is caused by single gene defects in any one of these factors, which are named for their complementation groups (A–W to date) [99,115]. Importantly, RAD51 and its mediators make up a significant portion of these complementation groups, including BRCA2 (FANCD1), PALB2 (FANCN), RAD51C (FANCO), RAD51 (FANCR), and XRCC2 (FANCU) [99]. The addition of RAD51, RAD51C, and XRCC2 to the FA family has been quite recent likely due to the availability and decreased cost of DNA sequencing. For example, recessive truncation mutations in XRCC2 identified a new FA group: FANCU [98]. While RAD51C is classified as a FANC complementation group, it does not share all of the phenotypes that present in other FA subtypes. Therefore, it is considered an FA-like syndrome [96,97]. Two distinct cases have been reported thus far that have interesting and significant differences. The first case had biallelic point mutations (R258H), whereas the second had two distinct mutant alleles, a point mutation (R312Q), and a splice variant [96,97]. While biallelic mutations can cause individuals to develop FA, heterozygous RAD51C FA carriers are predisposed to cancer [97]. It is interesting to note that all of the members of the PALB2–RAD51–RAD51C–BRCA2 complex are FA genes, as well as cancer-associated. While the function of this complex is unknown, it is tempting to speculate that perhaps there is a commonality among these proteins that has yet to be experimentally addressed. Since the RAD51 paralogs interact with each other to function in complexes and their stability is intimately intertwined, it is possible that more FA patients with mutations in the RAD51 mediators may be identified. The emerging role of RAD51C and potentially other RAD51 paralogs in ICL repair is still an emerging field of study, and we have yet to determine whether these proteins are playing roles upstream of canonical HR during ICL repair.

### 3.2. Cancers Associated with Defects in Homologous Recombination

While patients with biallelic mutations in RAD51 and its mediators have been identified in FA patients, monoallelic germline mutations in RAD51 mediators are correlated to predisposition to cancer [93,116]. This is thought to be frequently caused by a somatic loss of heterozygosity (LOH) event, where the second functional copy of the gene is deleted, resulting in genomic instability and cancer development [117,118]. Cancer-associated HR mutations are commonly found in BRCA1 and BRCA2, and can be both germline and somatic. BRCA1 and BRCA2, which were initially named as breast cancer susceptibility genes 1 and 2, are heavily associated with breast and ovarian cancers [116,117,119]. As such, these genes are routinely screened for in women and men with a family history of breast or ovarian cancer [119]. BRCA1 mutations account for about 16.3% of ovarian cancers, and BRCA2 mutations account for 6% of ovarian cancers [116]. BRCA1/2 mutations are somewhat less frequent in breast cancers, accounting for 5.5–6.1% of patients [119]. Interestingly, the overall mutation frequency of BRCA1/2 decreases with age in breast cancer patients, suggesting that deleterious mutations in BRCA1/2 are found in earlier-onset cancers [119]. Mutations in these genes are diverse, and range from single amino acid changes to the methylation silencing of promoters [116,119,120]. More recently, mutations in additional RAD51 mediators are being screened for in the clinic (BARD1, BRIP1, PALB2, RAD51C, RAD51D) [116,119,121]. Mutations in HR genes are particularly abundant in breast, ovarian, and endometrial cancers, but have also been found in other cancers such as pancreatic and colon cancers [116,122,123,124]. A number of the RAD51 mediators have now been added to the more comprehensive breast and ovarian cancer screening panels (i.e., PALB2, RAD51C, RAD51D, XRCC2). Due to the technical challenges in studying these proteins described above, the vast majority of the identified mutations are variants of unknown significance. However, hundreds of epidemiology studies have tried to correlate specific RAD51 paralog mutations with cancer predisposition using population studies. A HR-deficiency is most prevalent in ovarian carcinomas, and an HR-deficient mutational signature is found in approximately 20% of breast cancers [125]. In the most lethal type of ovarian cancer, high-grade serous carcinomas, up to 51% of tumors are HR-deficient from either inherited or somatic mutations or the promoter methylation [116,120,126]. It has been estimated that 3% of hereditary ovarian cancer patients have a mutation in RAD51C, whereas 5% have a mutation in RAD51D [116]. Standard of care for HR-deficient ovarian cancer patients includes aggressive surgery and a combination of platinum and taxane chemotherapy [116,127,128]. However, the five-year survival rate is only 30%, and within 12 months, 30–40% of patients relapse [128]. To remedy these startling statistics, it is essential to target these patients with targeted chemotherapy. There remains a critical need to identify all of the HR-deficient tumors to determine who will most benefit from the therapies that are used to currently treat BRCA1/2 patients.

### 3.3. Therapeutic Strategies for Homologous Recombination-Deficient Breast and Ovarian Cancers

New treatment strategies targeting DDR factors have shown promising success in clinical trials [126,129]. These therapies can effectively kill cancer cells with defects in DNA repair through synthetic lethality [126]. However, understanding which tumors will be vulnerable to a specific therapy is central to determining the efficacy of a drug for an individual tumor. This precision medicine approach requires detailed molecular analysis of variants of unknown significance to determine if the target gene is truly deleterious and a good candidate for targeted therapy. Synthetic lethality with the HR pathway has proved especially effective for BRCA-deficient cancers. Similarly, emerging evidence suggests these therapies will also be efficacious in targeting other HR gene defects. In addition to traditional chemotherapeutic drugs that induced DNA damage, HR-deficient tumors are currently being targeted with small molecule inhibitors of poly (ADP-ribose) polymerase (PARP), DDR signaling molecules, and NHEJ factors [118]. These inhibitors have had varying levels of success, and some have already been introduced to the clinic, while others remain in various stages of clinical testing, as described below.

## 4. Synthetic Lethality in BRCA and BRCA-Like Cancers

Synthetic lethality induces cell death by the loss of two somewhat redundant functions that alone would have been viable, such as the impairment of two DNA repair pathways that respond to the same DNA lesions [126]. Synthetic lethality can be achieved through numerous targeting strategies in cancer cells with a genetic loss-of-function mutation. Cancer cells typically lose some component of the DDR pathway that has enabled genomic instability, and has therefore been selected for, as it allows the tumor to grow [129,130,131]. This loss of function is specific to the tumor, and causes the tumor cells to have greater dependence on the remaining pathway(s) for survival relative to normal cells. An effective strategy in specifically targeting cancer cells for cell death takes advantage of DDR impairment in the tumor by pharmacologically impairing a complementary pathway that becomes essential for cell survival [129,130]. The protection of normal cells from this same lethality is an advantage for traditional chemotherapies that induce DNA damage in all cells [129,130].

As discussed above, BRCA-deficient cancer cells typically arise through a LOH event that enables tumor growth through an increased tolerance for genomic instability [118,129]. However, the surrounding cells still maintain one normal copy of the BRCA gene, and are therefore considered BRCA-proficient [118]. The tumor then relies on alternative repair pathways, while the surrounding tissue can still use HR. Therefore, a drug impairing only the HR alternative pathways will have little effect on normal cells, limiting its toxicity [129]. This difference between the two tissue types provides an opportunity for targeted therapy. By increasing the damage burden of the cell through classical chemotherapy and/or inhibiting back-up repair pathways (SSA or alternative NHEJ) through small molecules, the tumor cells can be specifically targeted [22]. This minimizes the impact on normal cells while effectively killing cancer cells [132]. This concept has produced a variety of therapies to treat BRCA and BRCA-like cancers [126]. It is interesting that HR-defective tumors are sensitive to PARP inhibitors, which were designed to target PARP1 and increase ssDNA breaks, which were thought to become DSBs in a replicative context [129]. New findings have suggested that PARP1 might also play a critical role during Okazaki fragment ligation during replication to facilitate repair [133]. Currently, HR-deficient cancers are primarily clinically specific to BRCA-deficient cancers; other HR factors are less well characterized in a clinical setting, but could benefit from the same strategies.

## 5. Chemotherapeutics Currently in the Clinic

Classically, ovarian cancers have been treated with platinum-based chemotherapeutics, and more recently, triple-negative breast cancer patients have increased progression-free survival using platinum therapy [134]. BRCA1/2-deficient tumors show increased sensitivity to platinum-based therapy, resulting in an improved overall survival [118,128]. Platinum-based drugs including cisplatin and carboplatin are effective inducers of interstrand and intrastrand crosslinks that require ICL and subsequently HR repair to resolve [113,134]. In BRCA1/2-deficient tumors, HR is incapable of resolving ICLs, and DSBs will be repaired by error-prone alternative pathways, leading to cell death in the tumor cells. The increased DNA damage is much more toxic to the HR-deficient tumor cells than to the surrounding HR-proficient cells. Similar to BRCA1/2-deficient tumors, emerging studies show similar platinum sensitivities for mutations in HR genes, including RAD51C and RAD51D [135,136]. While initially effective, subsequent platinum resistance often leads to the recurrence of the tumor [116]. In this scenario, ovarian cancer patients have now been approved for the use of PARP inhibitors (PARPi) [137].

PARP inhibitors are small molecule inhibitors that have recently been approved for platinum-sensitive relapsed and platinum-resistant ovarian cancers specifically harboring BRCA1 or BRCA2 mutations [129,132,137]. Toxicity from PARPi within the BRCA1/2-deficient tumor cells is much higher than in the surrounding BRCA1/2-proficient tissue. These findings suggest a therapy strategy for other HR-deficient tumors. The relationship between PARP and HR is complex, as PARP is a single-strand break repair factor. PARPi range in their ability to trap PARP on DNA through the competitive binding of the inhibitor, where NAD^+^ activates PARP, releasing it from the DNA [129,138]. However, when PARP dissociation from DNA is inhibited, it becomes trapped on DNA, and will block DNA replication [129]. This replication block generates stalled or collapsed replication forks that require HR for repair. It is possible that PARP inhibition may also function by preventing Okazaki fragment ligation, which would require HR repair for removal [133]. If HR is impaired, as occurs in BRCA-deficient tumor cells, more deleterious HR alternative pathways repair the damage, and lead to rampant genome instability and eventually cell death [129]. Currently, Olaparib and Rucaparib are two PARPi that have been approved for use in the United States [139]. These drugs show different levels of PARP trapping on the DNA where niraparib, olaparib, and rucaparib are medium trappers relative to other PARPi such as talazoparib and veliparib, which are stronger or weaker trappers [138]. The success of these drugs in platinum-resistant ovarian cancers holds promise for the expanded approval of a larger set of ovarian cancers, as well as breast cancers [129]. Currently, clinical trials are testing non-BRCA HR-deficient tumors that have other HR gene mutations, including RAD51C and RAD51D, which show PARPi response [140].

One of the major problems in treating ovarian cancers with PARPi is acquired resistance to the PARPi therapy due to a strong drive for reversion mutations, and therefore the restoration of HR. The first-line treatment of combined platinum and PARPi may be able to more effectively eliminate the tumor before resistance can arise. Recently, the acquisition of PARPi-resistance has been investigated in BRCA1-deficient cancer cells, and has also been observed in RAD51C and RAD51D patients [94]. In addition to reversion mutations restoring BRCA1 function directly, mutations in other genes that restore HR through alternative mechanisms have also been uncovered. For example, in BRCA-deficient tumors, the additional loss of 53BP1 or the shieldin genes rescues HR-deficiency by allowing the HR pathway to be engaged [14,15,16,17]. These results suggest that PARPi may be more effective when used in combination rather than as a monotherapy. In this scenario, a weaker PARP trapper or a reduced dose may be necessary to minimize cytotoxicity.

## 6. Summary

RAD51 filament formation is a central step in HR and is highly conserved throughout eukaryotes. RAD51 nucleoprotein filaments are tightly regulated by RAD51 mediator proteins, which serve to aid nucleation, elongation, stability, and disassembly. New roles for RAD51 mediators show they also facilitate filament flexibility and end capping, and have replication-specific functions. Importantly, mutations in the RAD51 paralogs are highly associated with hereditary breast and ovarian cancer predisposition, and more recently with several other cancers, including melanoma, colon, and pancreatic cancers. Despite the discovery of the human RAD51 paralogs three decades ago, few advances have been made in understanding their HR function. The RAD51 paralogs are challenging to study, because of the embryonic lethality observed in mouse knockout models, and their low protein abundance and insolubility. Currently, there is a critical need to understand the function of the wild-type RAD51 paralogs and determine how their mutations contribute to cancer predisposition. Here, we focused on the human RAD51 paralogs—RAD51B, C, D, XRCC2, 3, and SWSAP1—and explored their known and emerging functions, the challenges in further uncovering their roles, and their association with cancer predisposition. Future basic and clinical studies are required to uncover how the RAD51 mediators function to prevent cancer and develop targeted therapies toward the tumor-harboring mutations in these critical regulators.

## Figures and Tables

**Figure 1 genes-09-00629-f001:**
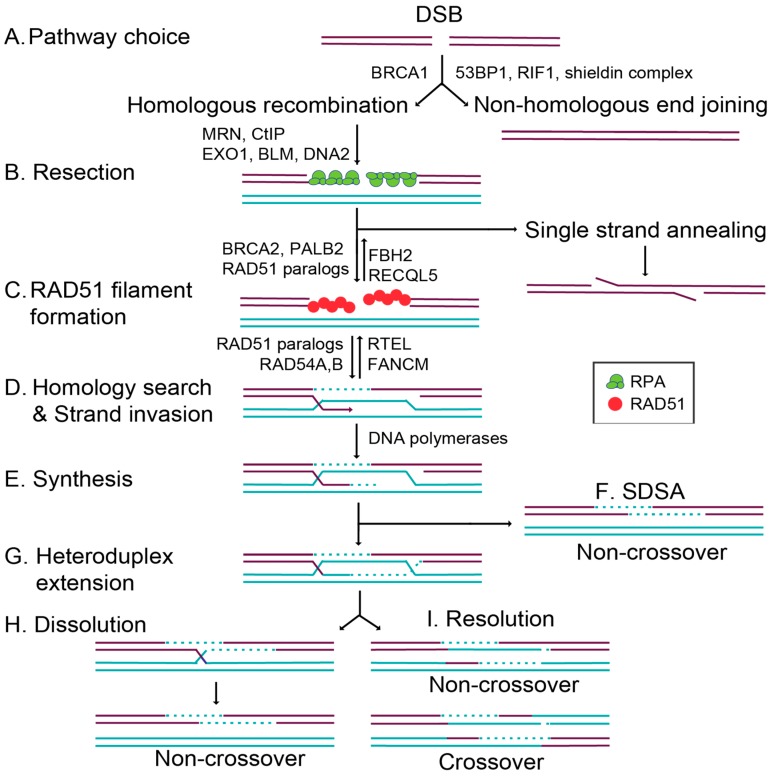
Schematic of Figure 1 double-strand break repair pathways and homologous recombination (HR). After the formation of a double-strand break (DSB; purple lines), cells can repair the damage through two primary mechanisms, HR using a homologous template (turquoise lines) or nonhomologous end joining (NHEJ). (**A**) Pathway choice between HR or NHEJ is mediated by BRCA1, which promotes HR and 53BP1, RIF1, and the shieldin complex, which promotes NHEJ. (**B**) Resection by the MRN (MRE11–RAD51–NBS1) complex, CtIP, EXO1, BLM, and DNA2 creates 3′ ssDNA overhangs, which are coated by the trimeric replication protein A (RPA) [20] complex (green circles). During canonical HR, RPA is displaced by RAD51 (red circles). Alternatively, RAD51-independent repair can occur through single-strand annealing, where complementary DNA sequences anneal, flap endonucleases cleave the overhang, and the DNA ends are ligated together. (**C**) RAD51 filament formation is regulated by the positive RAD51 regulators, BRCA2, PALB2, and the RAD51 paralogs. At the same time, RAD51 is negatively regulated by FBH2 and RECQL5. (**D**) RAD51-mediated homology search and strand invasion then occurs, and is regulated by the RAD51 paralogs and RAD54A,B. At the same time, RAD51-mediated D loops are negatively regulated by RTEL and FANCM. (**E**) The DNA polymerases then copy the missing information from the homologous template (shown in turquoise, a sister chromatid or a homologous chromosome). (**F**) During synthesis-dependent strand annealing (SDSA), the D loop is displaced, and the DNA is resolved into a noncrossover product. (**G**) If there is heteroduplex extension and a double Holliday junction formed by second-end capture, then these DNA intermediates can be resolved by dissolution or resolution. (**H**) Dissolution results in non-crossover products. (**I**) Resolution results in both crossover and non-crossover products.

**Figure 2 genes-09-00629-f002:**
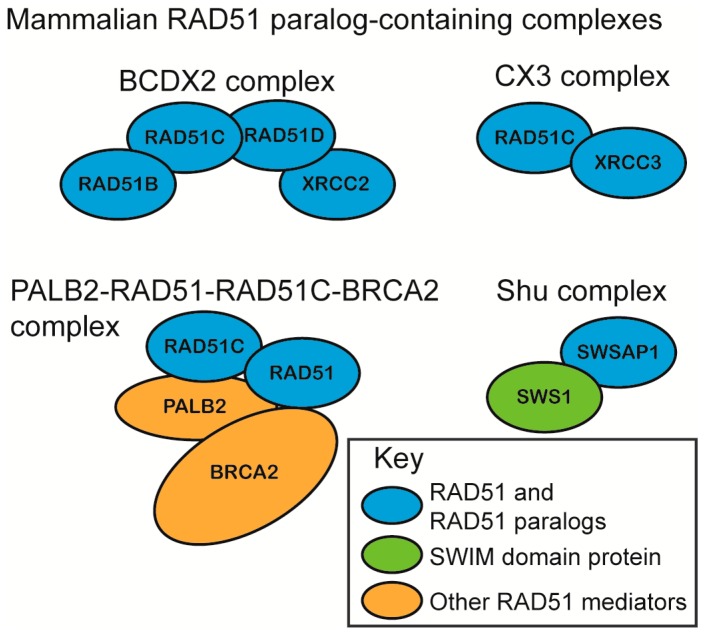
Mammalian RAD51 paralog-containing complexes. The BCDX2 complex is a RAD51 paralog heterotetramer consisting of RAD51B, RAD51C, RAD51D, and XRCC2. The CX3 complex is a RAD51 paralog heterodimer consisting of RAD51C and XRCC3. The PALB2–RAD51–RAD51C–BRCA2 complex consists of the RAD51 paralog, RAD51C, RAD51 itself, and two additional RAD51 mediator proteins, BRCA2 and PALB2. PALB2 acts as a scaffold in this complex by interacting with RAD51, RAD51C, and BRCA2. The Shu complex consists of a highly divergent RAD51 paralog, SWSAP1, and its binding partner SWS1. SWS1 is a member of the evolutionarily conserved Shu2/SWS1 family, which contains a SWIM domain, and interacts with RAD51 paralogs throughout eukaryotes. The SWIM domain is a zinc finger-like domain, CXC…X_n_…CXHXXA, where X is any amino acid. Blue circles indicate RAD51 or a RAD51 paralog, a green circle indicates a SWIM domain containing a Shu2/SWS1 protein family member, and an orange circle indicates an additional RAD51 mediator protein.

**Figure 3 genes-09-00629-f003:**
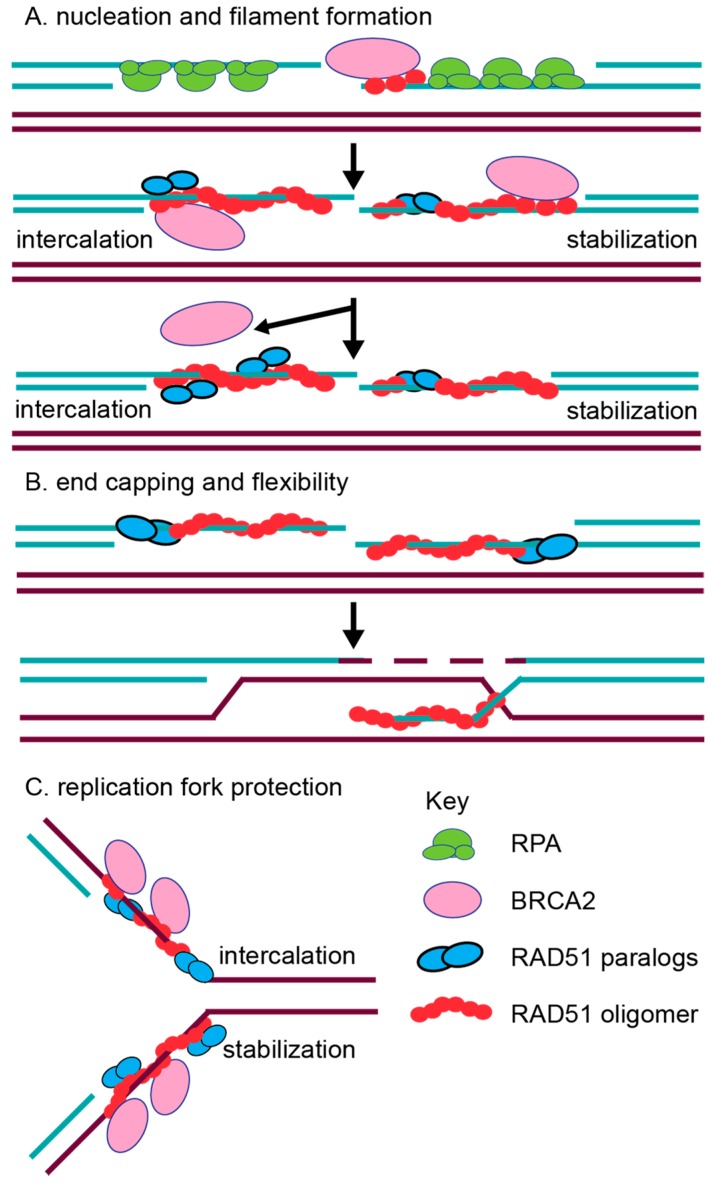
Proposed functions for the RAD51 paralogs. (**A**) Nucleation and filament formation. Traditionally, the RAD51 paralogs (blue circles) are thought to aid RAD51 (red circle) in filament formation with BRCA2 (pink circle), which nucleates RAD51 monomers onto ssDNA (turquoise lines). The RAD51 paralogs may act by intercalating into the RAD51 filament directly (shown on left) or by binding to the RAD51 filament to stabilize RAD51 (shown on right). Subsequently, BRCA2 is removed as the filament is stabilized and elongated. (**B**) RAD51 paralog function in RAD51 filament end capping and flexibility were recently proposed [39,40]. These functions would aid in RAD51 downstream activity to promote strand invasion. (**C**) Replication fork protection function has also been proposed for the RAD51 paralogs [85]. During fork protection, the RAD51 paralogs could potentially intercalate into (top strand) or stabilize (bottom strand) the RAD51 filament at a stalled replication fork.

**Table 1 genes-09-00629-t001:** Abbreviations Used.

BCDX2	RAD51B–RAD51C–RAD51D–XRCC2 complex
BC	RAD51B–RAD51C subcomplex
BER	Base excision repair
BIR	Break-induced replication
CHO	Chinese hamster ovary
CO	Crossover
CX3	RAD51C–XRCC3 complex
DDR	DNA damage response
DX2	RAD51D–XRCC2 subcomplex
dHJ	Double Holliday junction
DSB	Double-strand break
dsDNA	Double-stranded DNA
D-loop	Displacement loop
FA	Fanconi anemia
FANCD1	BRCA2
FANCN	PALB2
FANCO	RAD51C
FANCR	RAD51
FANCU	XRCC2
HR	Homologous recombination
ICL	Interstrand crosslink
IR	Ionizing radiation
LOH	Loss of heterozygosity
MEF	Mouse embryonic fibroblast
MMC	Mitomycin C
MMS	Methylmethane sulfonate
MRN	MRE11–RAD51–NBS1
NCO	Non-crossover
NHEJ	Non-homologous end joining
PARP	Poly (ADP-ribose) polymerase
PARPi	Poly (ADP-ribose) polymerase inhibitor
SCEs	Sister chromatid exchanges
SDSA	Synthesis-dependent strand annealing
ssDNA	Single-stranded DNA
**TIRF**	Total internal reflection fluorescence

**Table 2 genes-09-00629-t002:** RAD51 paralog knockout mice and derived mouse embryonic fibroblast (MEF) phenotypes.

RAD51 Paralog	Complex Member	Mouse Knockout	p53^−/−^ Rescue	MEF MMC Sensitivity	MEF MMS Sensitivity	Other Phenotypes
RAD51B	BCDX2	E7.5-8.5 [89]	partial [89]	NA	NA	NA
RAD51C	BCDX2, CX3, PALB2-RAD51- RAD51C-BRCA2	E8.5 [95]	partial [87]	2–3 fold (p53^−/−^) [87]	2–3 fold (p53^−/−^) [87]	↓SCEs [87] ↓IR-RAD51 foci (p53^−/−^) [87]
RAD51D	BCDX2	E9-E10 [88]	Partial [92]	17.6 fold (p53^−/−^) [92]	6.3 fold (p53^−/−^) [92]	↓SCEs (p53^−/−^) [92] ↓IR-RAD51 foci (p53^−/−^) [92]
XRCC2	BCDX2	E10.5-died at birth [86]	yes died 6d P/N [61]	4.5 fold (p53+) [91]	NA	↓SCEs [91] ↓IR-RAD51 foci [91]
XRCC3	CX3	NA	NA	NA	NA	NA
SWSAP1	Shu Complex (SWS1)	Viable/Infertile [84]	NA	NA	NA	NA

The phenotypes of RAD51 paralogs (RAD51B, RAD51C, RAD51D, XRCC2, XRCC3, and SWSAP1) are described, and references are indicated with superscript numbers. The complex where each RAD51 paralog is associated is indicated; the viability and lethality of the mouse knockout model is indicated, the degree of rescue by p53 deletion is shown, and mitomycin C (MMC) or methylmethane sulfonate (MMS) sensitivity from derived MEFs is also indicated. Not applicable is indicated by NA. Other phenotypes noted include sister chromatid exchanges (SCE), ionizing radiation-induced RAD51 foci (IR-RAD51), and a downward arrow (↓) indicates a reduction. P/N is post-natal.

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
