# Peer review of "RAD-ical New Insights into RAD51 Regulation"

_genes, 2018, doi:10.3390/genes9120629_

Reviewer 1 Report

In this review, the authors highlight the function of Rad51 and its paralogs in HR repair of DSBs and the association between defects in HR and cancer. Likewise, the authors provides an overview of the therapeutic strategies for HR deficient cancers based on the concept of synthetic lethality. This is an important area in tumor biology and cancer medicine, so the review topic is relevant.

This review paper has been very well written. I have few minor comments:

1) Line 312-313: RAD51KO/KO or RAD51 KO/KO. Please keep it consistent throughout the manuscript.

2) Table 1: please check the abbreviations appeared in the table. IR-RAD51 is not appearing in the table. Same is for downward arrow. Likewise, references should be cited for each Rad51 paralog in an additional column in the table.

3) Line 463: please check this sentence.

Author Response

We thank the reviewer for their comments and suggestions.

1) We have now fixed the text to have consistent RAD51KO/KO nomenclature.

2) We corrected the table to include a column that had been inadvertently deleted and included references.

3) We have altered Line 463 sentence to be clearer.

Reviewer 2 Report

This is an excellent review of the state of RAD51 paralog research and the challenges associated with unknowns in this protein family. The authors present clear reviews of the literature and excellent figures illustrating pathways. The only minor issue is that section 5 is not well linked with the rest of the review. It is unclear how knowledge of RAD51 paralogs impacts the current therapies or the PARPi treatments. 

The only other minor issue is inconsistent abbreviation use. Often terms like mitomycin are used in one section then abbreviated in another.  It may also be helpful to have an additional table linking the abbreviations of the paralogs, i.e., XRCC2 (FANCU), with proposed repair pathways. 

Author Response

We thank the reviewer for their insightful comments and suggestions.

We completely agree that knowledge of RAD51 paralogs does not impact current therapies or PARPi treatments.  We meant to convey that RAD51 paralog deficient tumors could benefit from the same treatment strategies used for BRCA-deficient tumors.  We softened the language where applicable.

We have also now included an New Table (new Table 1) with all the abbreviations used and checked the text to make sure that the abbreviations were only defined once in the text and in the legends for clarity.